# Nanovaccines: Innovative Advances from Design Strategies to Clinical Translation

**DOI:** 10.3390/vaccines13090900

**Published:** 2025-08-25

**Authors:** Jiuxiang He, Wen Xiao, Dong Hua, Minchi Liu, Hongxia Guo, Li Xu, Meiling Xiao, Yunsha Du, Jintao Li

**Affiliations:** Department of Biosafety, College of Basic Medicine, Army Medical University, Chongqing 400038, China; xiang18@tmmu.edu.cn (J.H.); xiaowen@tmmu.edu.cn (W.X.); huadong@tmmu.edu.cn (D.H.); minchiliu@tmmu.edu.cn (M.L.); hongxiaguo@tmmu.edu.cn (H.G.); xuli_2020@163.com (L.X.); 17318204382@163.com (M.X.); 13022354361@163.com (Y.D.)

**Keywords:** nanovaccines, targeted delivery, immunogenicity, cancer immunotherapy, infectious diseases

## Abstract

Nanovaccines have emerged as a transformative platform in immunotherapy, distinguished by their capabilities in targeted antigen delivery, enhanced immunogenicity, and multifunctional integration. By leveraging nanocarriers, these vaccines achieve precise antigen transport, improve immune activation efficiency, and enable synergistic functions such as antigen protection and adjuvant co-delivery. This review comprehensively explores the foundational design principles of nanovaccines, delves into the diversity of nanovaccine design strategies—including the selection of primary carrier materials, functionalization modification, synergistic delivery of immune adjuvants, and self-assembled nano-delivery systems—and highlights their applications in cancer immunotherapy, infectious disease and autoimmune diseases. Furthermore, it critically examines existing technical challenges and translational barriers, providing an integrative reference to guide future research and development in this dynamic field.

## 1. Introduction

The global burden of viral infectious diseases and the increase in the mortality rate of cancers pose a substantial threat to social stability and economic development. Vaccination remains the most effective tool for preventing and controlling infectious diseases and cancers, underscoring the critical need for safe and efficacious vaccines to safeguard global health and drive economic growth [1,2,3,4]. The COVID-19 pandemic in 2019 has further highlighted the indispensable role of advanced technologies, particularly nanotechnology, in accelerating vaccine development and enhancing their effectiveness [5].

Compared with conventional vaccines, nanotechnology-based vaccines (nanovaccines) offer distinct advantages, including improved antigen stability, enhanced delivery efficiency, and modulation of immune responses [6]. Traditional vaccines—such as live attenuated vaccines, inactivated vaccines, and subunit vaccines—each have inherent limitations [5]. Live attenuated vaccines, while immunogenic, carry the risk of reversion to pathogenic forms, whereas inactivated vaccines often elicit weaker immune responses, necessitating multiple booster doses. Although epitope-based subunit vaccines have emerged as safer alternatives, their limited immunogenicity typically induces only partial protective immunity and fails to elicit robust, long-lasting immune responses [7,8]. Moreover, conventional vaccines primarily stimulate humoral immunity through the generation of neutralizing antibodies, but are less effective in activating cellular immunity, which is essential for the complete clearance of intracellular pathogens. This constrains their utility in fully preventing or treating certain diseases.

Additionally, traditional vaccines face challenges related to stability and precise targeting. Being predominantly composed of biological macromolecules such as proteins or polysaccharides, they are susceptible to denaturation or degradation under suboptimal storage and transport conditions, compromising vaccine efficacy [9]. Their use of whole pathogens or broad antigenic components often results in limited specificity and suboptimal delivery to target immune cells, thereby diminishing immune activation. These limitations underscore the urgent need for innovative vaccine platforms.

Nanovaccines employ nanotechnology to encapsulate or conjugate antigens for targeted delivery to the immune cells. This facilitates more controlled, precise, and effective activation of host immune responses. Nanocarriers can present antigens on their surfaces to enhance uptake by antigen-presenting cells or encapsulate antigens within their structures to shield them from enzymatic degradation, prolong stability, and enable sustained antigen release, ultimately improving immunogenicity and protective outcomes. The advent of nanovaccine technology offers promising avenues for overcoming the drawbacks of traditional vaccines. Notably, cancer immunotherapy utilizing nanovaccines exemplifies the cutting edge of precision medicine [10]. In particular, mRNA vaccines play a significant role in the prevention and control of infectious diseases as well as tumor immunotherapy, owing to their advantages such as the ability to efficiently generate protective immune responses, relatively low side effects, and lower acquisition costs [11,12]. Positioned at the intersection of materials science and immunology, nanotechnology confers unique advantages—such as increased antigen stability, targeted delivery, controlled release, and synergistic adjuvanticity—that collectively enhance vaccine performance [13,14,15]. The potential applications of nanovaccines extend across diverse disease domains, including infectious diseases, cancers, and autoimmune disorders [4,16,17]. In infectious diseases, nanovaccines can bolster vaccine immunogenicity and protective efficacy, thereby aiding in the prevention and containment of outbreaks [18]. In oncology, they enable targeted delivery of tumor antigens, potentiate immune system activation, and impede tumor progression and metastasis. For autoimmune diseases, nanovaccines hold promise in modulating aberrant immune responses to mitigate disease symptoms and progression.

Despite their considerable advantages and encouraging preclinical outcomes, nanovaccines remain largely confined to the early stages of research, with significant hurdles yet to be addressed in the transition from animal studies to human clinical trials. This review aims to systematically examine the design strategies and core technologies underpinning nanovaccines, while elucidating their applications in cancer therapy, infectious disease prevention, and beyond. Furthermore, we critically analyze the prevailing technical challenges impeding clinical translation, with the goal of providing comprehensive insights to inform future research and development in this rapidly evolving field.

## 2. Strategies for the Design of Nanovaccines

### 2.1. Selection and Optimization of Primary Carrier Materials

In the development of nanovaccines, the choice and optimization of carrier materials are of paramount importance, as different materials exhibit distinct physicochemical properties and functional advantages (Figure 1). Nanoparticles couple antigens/adjuvants to nanocarriers via physisorption or chemisorption for targeted delivery and immune stimulation [19,20,21]. The stability of nanoparticles is a key challenge in vaccine development, impacting delivery efficiency, immunogenicity, and safety of the vaccine. Multiple factors—physical, chemical, and environmental—affect nanoparticle stability. By optimizing surface modification, the preparation process, storage conditions, and the use of stabilizers, the stability of nanoparticles can be effectively improved [22,23].

#### 2.1.1. Inorganic Nanoparticles

Inorganic nanoparticles—including gold nanoparticles, iron oxide nanoparticles, silica nanoparticles, and carbon nanotubes—have emerged as promising carriers in nanovaccine design due to their structural stability and favorable biocompatibility [24,25]. These materials facilitate antigen pre-conjugation and targeted delivery. Their inherent stability enables prolonged bioactivity in vivo, while their ability to enhance permeability and retention (EPR) effects allows preferential accumulation at pathological sites, such as tumors, thereby improving delivery efficiency [26]. However, the EPR effect varies greatly in different lesion types, such as solid tumor heterogeneity, which is not universal. It is more of an auxiliary delivery method, needing to be combined with the active targeting strategy to achieve better effects. Additionally, inorganic nanomaterials can serve as scaffolds to integrate other biomaterials, yielding composite delivery systems with superior mechanical strength and functional versatility.

Beyond their role as carriers, certain inorganic nanomaterials possess intrinsic immunostimulatory properties, contributing to the activation and enhancement of host immune responses [27,28]. However, the stability of inorganic nanomaterials is a double-edged sword. Materials such as carbon nanotubes and colloidal gold often exhibit limited solubility and poor biodegradability, leading to prolonged retention in the body. This can result in adverse outcomes, including inflammation, fibrosis, or other toxic side effects [25].

Therefore, the design and development of biodegradable inorganic nanomaterials are imperative to mitigate long-term safety concerns and to enable their broader application in biomedical and vaccine delivery contexts.

#### 2.1.2. Polymeric Nanoparticles

Polymeric nanoparticles represent a versatile vaccine delivery platform in which antigens and adjuvants are encapsulated within a biocompatible and biodegradable polymer matrix. Compared with traditional vaccine delivery systems, polymeric nanoparticles offer several advantages, including enhanced antigen stability, protection against enzymatic degradation, controlled release kinetics, and the ability to target specific cells or tissues [29].

It has been demonstrated that the chemical structure and physicochemical properties of polymeric nanomaterials can be finely tuned by modifying synthesis conditions and formulation parameters. This allows precise control over particle size, morphology, and surface characteristics, thereby improving both the stability and the targeting efficiency of nanovaccines. Common polymeric materials employed for nanovaccine fabrication include poly(lactic-co-glycolic acid) (PLGA), chitosan, polylactic acid (PLA), and polyethylene glycol (PEG), all of which exhibit excellent biocompatibility, biodegradability, and modifiable surface properties, making them highly attractive carriers for vaccine delivery [30,31,32].

Among these, PLGA is the most extensively studied polymer for biomedical applications. As a biodegradable copolymer, PLGA undergoes hydrolysis of its ester bonds, leading to safe metabolic by-products, and has been approved by the U.S. Food and Drug Administration (FDA) for human clinical use [33]. When antigens are encapsulated within PLGA nanoparticles, they are gradually released as the polymer matrix degrades in vivo. For example, studies on influenza vaccines have demonstrated that PLGA nanoparticles encapsulating influenza virus antigens can induce sustained antibody and cellular immune responses in murine models, while providing cross-protection against different influenza subtypes [34].

This sustained-release capability maintains an effective antigen concentration over extended periods, continuously stimulating the immune system and thereby enhancing immunogenicity. In addition to their favorable release profiles, PLGA nanoparticles are highly biocompatible and do not induce long-term toxicity or adverse reactions. Furthermore, the synthesis process and formulation parameters allow precise modulation of their particle size and surface properties, facilitating optimized biodistribution and immune activation [35].

#### 2.1.3. Lipid Nanoparticles

Lipid nanoparticles have attracted considerable attention as vaccine delivery systems owing to their excellent biocompatibility, low intrinsic toxicity, and controlled release properties. They are capable of encapsulating antigens and adjuvants, protecting them from enzymatic degradation, and facilitating targeted delivery to specific cells or tissues. Among the various nanocarriers, lipid nanoparticles offer notable advantages in terms of encapsulation efficiency and scalability of manufacturing, making them particularly suitable for large-scale vaccine production [36]. As endogenous-like components, lipid nanoparticles also help reduce immune rejection and enhance the stability and efficacy of vaccines.

Common lipid-based nanocarriers include liposomes, nanoemulsions (such as nanostructured lipid carriers, NLCs), and solid lipid nanoparticles (SLNs). These systems can encapsulate both hydrophilic and hydrophobic antigens, improving the solubility of poorly water-soluble molecules and enabling delivery through fusion with target cell membranes. Liposomes, for instance, are spherical vesicles composed of a phospholipid bilayer surrounding an aqueous core, which protects encapsulated antigens from gastrointestinal degradation and supports sustained release [37]. However, despite their versatility, liposomes often exhibit limited loading capacities, particularly for large biomolecules such as mRNA or proteins [38]. Moreover, mRNA vaccines employing lipid nanoparticles require ultra-low temperature storage and distribution, complicating logistics and widespread deployment.

#### 2.1.4. Biomimetic Nanoparticles

Biomimetic nanoparticles, designed to emulate the structure and functions of natural cells, have emerged as promising platforms to enhance vaccine biocompatibility and elicit cell-like interactions [39]. Due to their complex biological functionalities, these materials can efficiently deliver antigens and potentiate immune responses [40]. Examples include virus-like particles (VLPs), bacterial outer membrane vesicles (OMVs), erythrocyte membranes, and dendritic cell membranes, all of which mimic natural cellular properties to improve targeting and immunogenicity [3].

Among these, OMVs—nanoscale vesicles naturally secreted by Gram-negative bacteria—have gained traction in personalized cancer vaccine strategies. OMVs are primarily composed of outer membrane proteins and lipopolysaccharides, which inherently display pathogen-associated molecular patterns (PAMPs). These features enable OMVs to activate various immune cells, such as macrophages and dendritic cells, by triggering innate immune pathways and initiating robust adaptive responses. Because of their intrinsic immunostimulatory properties, OMVs can act as natural adjuvants to enhance antigen-specific immunity [41]. Genetic engineering further allows OMVs to be modified to present specific tumor or pathogen antigens, thereby constructing targeted vaccine vectors. In studies on *Neisseria meningitidis*, OMV-based vaccines successfully induced strong immune responses, generating specific antibodies that conferred protection against meningitis [42].

As a frontier in nanomedicine, biomimetic nanotechnology has also been widely explored for cancer therapy to improve efficacy while minimizing systemic toxicity. The tunable size, morphology, and surface characteristics of biomimetic nanomaterials facilitate the encapsulation of hydrophobic drugs, controlled release, improved pharmacokinetics, and efficient tissue penetration. These nanoparticles can traverse narrow capillaries, enabling passive targeting through enhanced permeability and retention (EPR) effects, or active targeting via ligand-mediated interactions with specific cellular receptors. Similar to lipid nanomaterials, biomimetic systems can be surface-functionalized to specifically target cancer cells or the tumor microenvironment (TME), thereby enhancing delivery precision and therapeutic outcomes [43].

Nonetheless, the design and fabrication of biomimetic nanomaterials are often complex, involving multi-step processes and sophisticated technologies that necessitate specialized equipment. This results in high production costs and poses significant challenges to scalability. Currently, these materials are primarily confined to laboratory research, with large-scale, reproducible manufacturing remaining difficult to achieve [44].

### 2.2. Functionalization to Enhance Targeting

Functionalization strategies have opened new avenues to improve the targeting capabilities of nanovaccines. Active targeting is primarily achieved by modifying the surfaces of nanovaccines with antibodies, peptides, or other ligands that can specifically bind to receptors on target cells or tissues [45]. Such surface ligand modifications, along with advanced designs like pH-responsive membrane fusion systems, enhance the precision with which nanovaccines recognize and act on specific biological targets [46,47].

By decorating the surfaces of nanovaccines with specific ligands, it is possible to achieve active targeting of dendritic cells (DCs) and other key immune cells. For instance, the modification of nanocarriers with polyethylene glycol (PEG) not only reduces nonspecific protein adsorption but also improves the overall targeting efficiency of the nanovaccines [48,49]. Incorporating ligands that recognize DC-specific surface receptors, such as C-type lectin receptors (CLRs), including DEC-205, BDCA-2 and dendritic cell inhibitory receptor 2 (DCIR2), facilitates the direct presentation of antigens to DCs [50,51]. Nanovaccines modified in this way can be internalized by various DC subsets, thereby enhancing antigen delivery and immune activation [52,53].

For cancer nanovaccines, functionalization can also leverage physiological cues—such as local pH or enzymatic activity—to trigger controlled drug release, protecting antigens and adjuvants from premature degradation and improving both stability and therapeutic efficacy [15]. An illustrative example is the use of vesicular stomatitis virus glycoprotein (VSVG), a key structural protein on the viral envelope, which is mainly responsible for viral attachment, fusion, and formation of viral particles. It possesses unique pH-sensitive membrane fusion properties. At physiological pH (~7.4), VSVG remains in a stable conformation. However, upon exposure to acidic environments, such as those found in the tumor microenvironment (pH 6.5–6.8) or in endosomes (pH 5.0–6.0), VSVG undergoes conformational changes that promote membrane fusion [54].

This pH-responsive fusion design enables nanovaccines to achieve active targeting of cancer cells, overcoming the limitations of traditional nanocarriers that depend solely on passive targeting mechanisms like enhanced permeability and retention (EPR) effects. Direct intracellular release of antigens within cancer cells further augments cellular immune responses, particularly the activation of cytotoxic T lymphocytes (CTLs), thereby strengthening antitumor effects. In preclinical models, nanoparticles functionalized with VSVG and loaded with cancer antigens have demonstrated enhanced antigen presentation, robust anticancer immune responses, and significant tumor growth inhibition [55].

### 2.3. Synergistic Delivery of Immune Adjuvants

Nanoparticles themselves can function as adjuvants in cancer nanovaccines, amplifying immune responses and enhancing therapeutic efficacy. Self-adjuvanted nanovaccines represent an innovative approach that integrates antigen delivery with intrinsic immune-stimulating properties, simplifying vaccine composition while potentiating immunogenicity. However, despite their promising potential in cancer therapy and the prevention of immune-mediated diseases, these systems face notable challenges. The interactions between nanoparticles and immune cells involve multiple signaling pathways and diverse cell types, making the underlying mechanisms of action complex and not yet fully elucidated [56]. Furthermore, immunostimulatory nanoparticles can inadvertently trigger adverse immunotoxic effects, such as activation of the complement cascade and antibody responses upon entry into the circulation, potentially leading to inflammation or allergic reactions [57].

For complex diseases like cancer and emerging infectious diseases—where the immune system often exhibits dysfunction or evasion mechanisms—co-delivery of multiple adjuvants can synergistically activate complementary immune pathways. Combining STING agonists with TLR ligands, for instance, has been shown to robustly amplify immune responses by engaging distinct immunological signaling axes [58,59]. STING (stimulator of interferon genes), a pattern recognition receptor (PRR) that recognizes pathogen-associated molecular patterns (PAMPs) in cells, is an endogenous sensor in the endoplasmic reticulum for cytosolic DNA. Once activated, it initiates signaling cascades, resulting in potent type I interferon-mediated immune responses. Given its central role in antitumor, antiviral, and antimicrobial immunity, numerous natural and synthetic STING agonists have been employed in advanced vaccine formulations [60].

In models of melanoma, colorectal cancer, and HPV-associated cancers, nanovaccines incorporating STING agonists have effectively inhibited tumor growth, offering a promising, safe, and potent strategy for cancer immunotherapy [61]. Moreover, co-encapsulation of peptide neoantigens with optimized combinations of STING and TLR4 agonists into nanocarriers significantly enhances lymph node accumulation and uptake by antigen-presenting cells, and promotes robust antigen-specific CD8+ T cell responses, thereby restraining tumor progression [62]. This synergistic strategy not only improves vaccine stability but also augments the uptake, processing, and presentation of antigens by immune cells, enabling more sustained and effective immune responses [63].

Nevertheless, certain STING agonists, such as cyclic dinucleotides (CDNs), face limitations due to rapid diffusion from injection sites, reducing local bioactivity and provoking systemic inflammation. Encapsulation within nanoparticles mitigates these issues by facilitating targeted delivery to DCs in draining lymph nodes while limiting systemic exposure. This co-delivery approach enhances immune cell activation while reducing the risk of systemic inflammation, thereby improving both the efficacy and safety profile of the vaccine [64].

### 2.4. Self-Assembled Nano-Delivery Systems

Nanoparticle-based vaccines with an ideal size ranging from 20 to 100 nm, which mimic the size and structural presentation of natural viruses, enhance the recognition and uptake by immune cells (such as dendritic cells), providing an effective platform for triggering a strong immune response against infectious diseases [65]. Self-assembled nanovaccines are an innovative approach wherein nanomaterials with inherent self-assembly capabilities construct stable nanostructures incorporating antigens and adjuvants. These carrier-free systems avoid traditional delivery vectors, maintaining high antigen density while minimizing the toxicity risks associated with synthetic carriers [66,67]. This makes self-assembled nanovaccines a highly attractive strategy for the efficient delivery of synthetic antigens.

Virus-like particles (VLPs), composed of one or more self-assembling molecules that mimic the morphology and size of viruses but lack genetic material, represent a prime example. VLPs are highly immunogenic, engaging immune pathways distinct from those triggered by conventional inactivated virus vaccines, and can simultaneously activate humoral and cellular immune responses [68,69]. The successful commercialization of VLP-based vaccines for human papillomavirus (HPV), hepatitis B virus, and malaria underscores their promise. Ongoing studies continue to explore their application in vaccines targeting other infectious diseases and cancers [70,71].

Beyond virus-derived assemblies, various self-assembling proteins such as ferritin, mi3, E2 protein, and encapsulin proteins have been widely used in nanovaccine development [72,73,74,75]. These proteins can be engineered to present antigens through chemical modification or genetic fusion. Ferritin, for instance, is a naturally self-assembling protein composed of 24 subunits forming a cage-like structure with excellent biocompatibility and modifiability. It has been employed in vaccine research against a range of pathogens, including SARS-CoV-2, influenza virus, Zika virus, porcine rotavirus, and porcine reproductive and respiratory syndrome virus, eliciting robust protective immune responses with high safety profiles [76,77,78].

Additionally, self-assembling peptides that organize into nanofibers, nanotubes, nanoribbons, or hydrogels present significant potential for vaccine applications. These structures can promote cell adhesion, provide epitope recognition sites, and facilitate antigen presentation and release [79,80]. For example, the Q11 peptide can self-assemble into nanofibers under physiological conditions, and when conjugated with the highly conserved M2e epitope of influenza virus, forms the nanoparticle M2e-Q11 in isotonic solutions. This nanovaccine effectively induces protective antibodies against multiple influenza subtypes in mice without requiring additional adjuvants [81].

Overall, nanovaccines encompass a diverse array of delivery vectors, each offering unique advantages and facing distinct limitations (Table 1). The careful selection of appropriate nanocarriers is critical to advancing research and facilitating clinical translation, particularly in cancer immunotherapy. Comprehensive comparative analyses considering stability, efficacy, cost, manufacturability, and scalability can guide researchers and clinicians in making informed choices [3]. Furthermore, integrating complementary delivery strategies or optimizing hybrid systems may provide innovative solutions to overcome individual shortcomings, thereby enhancing the overall efficacy and safety of nanovaccine-based immunotherapies [10]. In cancer vaccine development, rigorous assessment of the specific benefits and drawbacks of each nanocarrier system is essential to ensure precise, effective, and safe immune activation.

## 3. Breakthroughs in the Field of Application

### 3.1. Cancer Immunotherapy

Vaccination plays an indispensable role in the prevention and treatment of diseases, and cancer nanovaccines have recently garnered considerable attention due to their capacity to elicit robust anti-tumor immune responses. Cancer vaccines, critical tools within immunotherapy, can be broadly categorized into prophylactic and therapeutic vaccines. Prophylactic vaccines, such as those targeting human papillomavirus (HPV) and hepatitis B virus (HBV), reduce the incidence of virus-associated malignancies by priming the immune system [82]. In contrast, therapeutic cancer vaccines are designed to stimulate dendritic cells (DCs) and cytotoxic T lymphocytes (CTLs), thereby establishing durable anti-cancer immunity [83]. An ideal cancer vaccine should induce potent, specific, and long-lasting immune responses that selectively target malignant cells. To achieve this, it is essential to incorporate tumor-specific antigens or neoantigens that are absent in normal tissues, thereby minimizing the risk of autoimmunity while concentrating the immune attack on cancer cells. Nevertheless, the low immunogenicity of most tumor antigens, coupled with tumor-mediated immunosuppressive mechanisms, poses significant challenges to the efficacy of cancer vaccines [25].

Recent advances in nanomaterials and cancer immunotherapy have facilitated the development of cancer nanovaccines, which have demonstrated promising potential in the management of melanoma, hepatocellular carcinoma, and triple-negative breast cancer [84]. These nanovaccines typically encapsulate or surface-load tumor-associated antigens, enhancing their stability and recognition while improving lymph node trafficking. Owing to their nanoscale dimensions, these vaccines can efficiently target lymphatic system and be taken up by immune cells, eliciting sustained anti-tumor immune responses [85,86,87,88].

Successful induction of adaptive anti-tumor immunity relies on three key elements: antigen, adjuvant, and an immune-supportive tumor microenvironment (TME) [89]. The TME—comprising tumor cells, immune infiltrates, stromal cells, fibroblasts, and an extracellular matrix—plays a pivotal role in tumor initiation, proliferation, invasion, metastasis, and therapeutic resistance [90]. Consequently, modulation of the TME represents an emerging strategy for cancer treatment. By targeted delivery of immunomodulatory components to the TME, the nanovaccine reverses its immunosuppressive state, destroys tumor immune escape, and amplifies the overall effect of anti-tumor immune response. For example, combining photothermal therapy with bacterial outer membrane vesicles (OMVs) modified by maleimide (OMV-Mal) has shown efficacy in reversing immunosuppressive microenvironments. Photothermal therapy locally heats tumor tissues via photoactivators, promoting cancer cell death and releasing tumor antigens. OMV-Mal captures these antigens, enhancing their presentation to immune cells. Given their intrinsic immunostimulatory properties, OMVs can further activate innate and adaptive responses and modulate the TME, generating a synergistic effect that not only eradicates tumor cells but also diminishes recurrence risk [91].

Moreover, combining nanovaccines with immune checkpoint inhibitors (ICIs) such as anti-PD-1/PD-L1 antibodies has demonstrated synergistic benefits. While nanovaccines increase the pool of tumor-specific T cells, ICIs block inhibitory pathways, enabling these T cells to function effectively within the TME [92]. This combinatorial strategy substantially amplifies anti-tumor immunity, offering promising therapeutic avenues, particularly for patients who do not respond adequately to monotherapies.

### 3.2. Prevention and Control of Infectious Diseases

The COVID-19 pandemic has underscored the transformative potential of nanotechnology in vaccine development. Nanovaccines have overcome many of the inherent limitations of traditional vaccines by facilitating targeted antigen delivery, enhancing antigen presentation, and enabling controlled release, thereby improving immunogenicity and vaccine stability [93,94].

Among the most notable achievements is the development of mRNA vaccines, which leverage lipid nanoparticles (LNPs) to deliver genetic instructions encoding viral antigens. These vaccines have demonstrated remarkable efficacy against SARS-CoV-2 and its variants [95]. For instance, the Pfizer/BioNTech and Moderna vaccines use LNPs to encapsulate mRNA encoding the spike protein, prompting host cells to express this antigen and thereby eliciting robust antibody and T cell responses. Clinical trials have reported efficacy rates as high as 95% alongside favorable safety profiles [96]. Furthermore, the modular and rapid production of mRNA-LNP platforms offers unparalleled adaptability for combating emerging pathogens [97].

Beyond COVID-19, nanovaccines are advancing the development of vaccines against other formidable infectious diseases. In HIV vaccine research, nanoparticle platforms capable of multivalent antigen display mimic viral structures and enhance B cell activation, broadening and intensifying immune responses. For example, presenting stabilized HIV Env gp140 trimers on nanoparticles can induce broadly neutralizing antibodies, a key milestone in HIV vaccine efforts [98,99,100].

Nanotechnology also holds promise in veterinary vaccines, which are critical for controlling zoonotic diseases and improving livestock productivity [2,101]. Biodegradable polymeric nanoparticles such as PLGA and chitosan have been employed to develop vaccines against pathogens like African swine fever virus (ASFV) and porcine reproductive and respiratory syndrome virus (PRRSV) [102]. For instance, mannose and CpG-modified PLGA nanoparticles loaded with ASFV antigens enhanced both humoral and cellular immunity [1]. Similarly, ferritin-based self-assembled nanoparticles displaying PRRSV epitopes elicited robust T cell responses in pigs, showcasing the versatility of nanoplatforms in veterinary immunology.

### 3.3. Autoimmune Diseases and Allergies

Nanocarriers also offer innovative strategies for managing autoimmune diseases and allergic disorders by delivering immunomodulatory agents or inducing antigen-specific tolerance [103,104]. Autoimmune diseases arise from aberrant immune attacks on self-tissues, and precisely delivering immunoregulatory molecules—such as IDO inducers or PD-L1 mRNA—to specific immune cells can restore immune homeostasis. The immune regulation of nanovaccines may involve multiple types of immune responses. In addition to allergic responses, attention should be paid to the potential effects on other immune processes such as innate immune signaling pathways and T cell subset balance.

For example, Wang and Li’s group at the University of Science and Technology of China developed low-immunogenic LNPs optimized for delivering PD-L1 mRNA to antigen-presenting cells (APCs) [105]. In models of rheumatoid arthritis, these engineered APCs expressed PD-L1, selectively inhibiting hyperactive T cells while expanding regulatory T cells (Tregs), thereby markedly attenuating disease progression [106]. Similarly, nanocarriers have been employed to deliver modulators to dendritic cells, suppressing their maturation and subsequent T cell activation, which interrupts the initiation of allergic responses. Nanocarriers can also inhibit Th2 polarization and cytokine secretion (e.g., IL-4, IL-5, IL-13) while promoting Treg differentiation, collectively mitigating allergic symptoms [107,108].

Thus, by enabling precise delivery and minimizing systemic side effects, nanovaccine platforms and nanocarriers offer powerful new modalities for the treatment of autoimmune and allergic diseases.

On the whole, nanovaccines have been useful in breakthroughs in cancer immunotherapy, prevention and control of infectious diseases, and autoimmune diseases. With targeted delivery, efficient immune activation, and customizability, they have demonstrated broad clinical prospects, and numerous formulations have progressed to clinical trials (Table 2). The trial details outlined in Table 2 are sourced from ClinicalTrials.gov, a reliable and extensive database of clinical studies.

## 4. Key Challenges in the Clinical Translation

Despite remarkable preclinical advances, the clinical translation of nanovaccines remains fraught with significant challenges and unresolved questions (Figure 2). Critical factors such as nanoparticle composition, assembly methods, surface chemistry, ligand decoration, mechanical rigidity, and surface charge profoundly influence their biological performance and safety profiles. The inherent complexity of these composite nanostructures complicates thorough toxicological evaluation.

### 4.1. Safety Issues

The expanding biomedical application of nanomaterials necessitates rigorous assessment of their potential biological toxicity, with particular concern regarding the long-term metabolic fate of inorganic nanoparticles.

#### 4.1.1. Potential Toxicity

Upon systemic administration, nanoparticles may be perceived as foreign by the immune system, triggering activation of immune cells and secretion of pro-inflammatory cytokines including tumor necrosis factor-α (TNF-α), interleukin-1β (IL-1β), and interleukin-6 (IL-6). Sustained inflammatory responses may inflict chronic tissue damage, impairing organ function [110]. Additionally, nanoparticles can induce reactive oxygen species (ROS) production via multiple pathways, generating oxidative stress that damages cellular macromolecules and disrupts redox homeostasis, potentially culminating in apoptosis or necrosis [111]. Lysosomal accumulation of nanoparticles may cause membrane destabilization, releasing acidic hydrolases and nanoparticles that compromise intracellular organelles and metabolism [112]. Moreover, nanoparticles may interact with intracellular biomolecules, altering gene expression by binding DNA or disrupting protein function and signaling pathways, thereby impairing normal cellular activities [113].

#### 4.1.2. Long-Term Metabolic Risks

Nanoparticles possess unique physicochemical properties distinct from endogenous biomolecules, complicating their metabolic processing and excretion. Many nanoparticles resist enzymatic degradation and evade clearance via conventional renal or biliary routes, resulting in prolonged in vivo persistence and cumulative toxicity [114,115]. Chronic retention raises concerns of carcinogenicity, reproductive toxicity, and genotoxicity. Al-though data remain limited, nanoparticles may provoke genetic mutations and chromosomal aberrations, and disrupt cell cycle regulation, potentially increasing cancer risk. They may also impair germ cell development and function, compromising reproductive health. To ensure safe clinical application, comprehensive mechanistic studies on nanotoxicity, establishment of standardized safety evaluation protocols, and stringent regulation of nanoparticle manufacture, use, and disposal are imperative.

### 4.2. Large-Scale Production and Stability

Successful therapeutic efficacy in animal models does not guarantee clinical translatability. Scaling up production of nanovaccines with consistent quality and stability remains a formidable hurdle. High labor and material costs, coupled with complex multi-step manufacturing protocols, impede reproducibility. Transitioning formulations from preclinical models to humans often necessitates optimization or redesign of manufacturing processes, which demands significant investment and innovation, frequently stalling promising candidates [116,117].

Specific challenges arise in the production of biologically derived components such as bacterial outer membrane vesicles (OMVs) and virus-like particles (VLPs). OMV yield and composition are influenced by variable bacterial culture conditions (media, temperature, pH, dissolved oxygen) and genetic stability, contributing to batch-to-batch heterogeneity that complicates purification and quality control. The high cost of specialized equipment, reagents, and labor further constrains large-scale OMV production [118,119].

VLP manufacturing depends critically on consistent expression and quality of viral structural proteins. Variability in gene expression, cell metabolism, and culture conditions can alter protein yield and conformation, affecting VLP self-assembly efficiency and product quality [120]. Environmental factors such as temperature, pH, and ionic strength influence VLP morphology, size distribution, and immunogenicity, necessitating precise process control. Achieving reproducible control over these parameters requires extensive process development and costly instrumentation, significantly escalating research and production expenses.

### 4.3. Balancing Immunogenicity and Targeting Efficiency

A pivotal challenge in nanovaccine development is achieving an optimal balance between potent immunogenicity and targeted delivery while avoiding adverse immune overactivation. Targeted delivery is one of the core objectives of nanovaccines. It not only relies on the inherent uptake of nanoparticles by myeloid cells, but also aims to accurately target functional myeloid cells (such as dendritic cells) through strategies such as ligand modification to improve the specificity and efficiency of immune responses. A cytokine storm—an uncontrolled systemic inflammatory response characterized by excessive release of cytokines like TNF-α and IL-6—poses a severe risk, potentially causing multi-organ damage and mortality.

Nanomaterial physicochemical properties (size, shape, surface charge, composition) affect their interaction with the immune system. The large surface area often leads to protein corona formation, which can modify nanovaccine immunogenicity and elevate inflammation risk [121]. While nanovaccines activate antigen-presenting cells and downstream T and B lymphocytes, improperly designed formulations may provoke excessive or prolonged immune activation. Adjuvants further augment immunogenicity but, if misused or overdosed, may precipitate cytokine storms.

Optimizing delivery platforms, formulations, dosage forms, and administration routes is critical to enhancing efficacy, safety, stability, and patient compliance, while facilitating large-scale immunization logistics [122].

## 5. Perspectives

Looking forward, nanovaccine technology is poised to evolve along multiple dimensions—intelligent design, interdisciplinary integration, novel delivery methods, and universal protection—poised to revolutionize vaccine science.

### 5.1. Intelligent Design Upgrades

Next-generation “smart” nanovaccines will feature stimulus-responsive release mechanisms tuned to the tumor microenvironment’s pH gradients, aberrant protease activity, and oxidative stress. For instance, multi-stage nanosystems can sequentially release antigens in acidic tumor tissues, followed by adjuvant activation via cancer-specific enzymes, amplifying immune responses in a spatiotemporal manner.

### 5.2. Multidisciplinary Integration and Innovation

Continued refinement of mRNA vaccine platforms and nanocarrier materials science will yield safer, more effective delivery systems. Artificial intelligence-driven machine learning models will accelerate in silico screening and optimization of mRNA sequences and carrier formulations. Additionally, integration with emerging gene-editing technologies promises expanded therapeutic capabilities in disease prevention and treatment.

### 5.3. Needle-Free Delivery and Mucosal Immunity Expansion

Needle-free administration, such as intranasal spray nanovaccines, will gain prominence, emphasizing enhanced mucosal adhesion, permeability, and immune activation to broaden protective coverage. Exploration of oral and rectal nanovaccine formulations will offer convenient, non-invasive options for infectious disease control and therapeutic interventions.

### 5.4. Breakthroughs in Universal Vaccines

Bioinformatics and structural biology will deepen identification of conserved epitopes across cancers and pathogens, facilitating universal vaccine development. Cancer vaccines targeting pan-cancer antigenic signatures and broad-spectrum infectious disease vaccines capable of covering multiple variants will mitigate immune escape and improve preparedness for global health crises.

## 6. Conclusions

Nanovaccines, at the intersection of materials science and immunology, have achieved significant milestones in both fundamental research and clinical translation. Technologies such as nanocarriers and functional surface modifications have surmounted traditional vaccine limitations, as exemplified by the global deployment of COVID-19 mRNA vaccines and emerging personalized cancer nanovaccines. As a next-generation vaccine platform, nanovaccines hold transformative potential for infectious disease prevention, cancer therapy, and immunomodulatory interventions. Nonetheless, challenges including nanotoxicity, manufacturing scalability, and immune response calibration must be addressed. Advancing interdisciplinary collaboration, process optimization, and regulatory frameworks will be pivotal to realizing the clinical promise of nanovaccines, ultimately enhancing global public health and personalized medicine.

## Figures and Tables

**Figure 1 vaccines-13-00900-f001:**
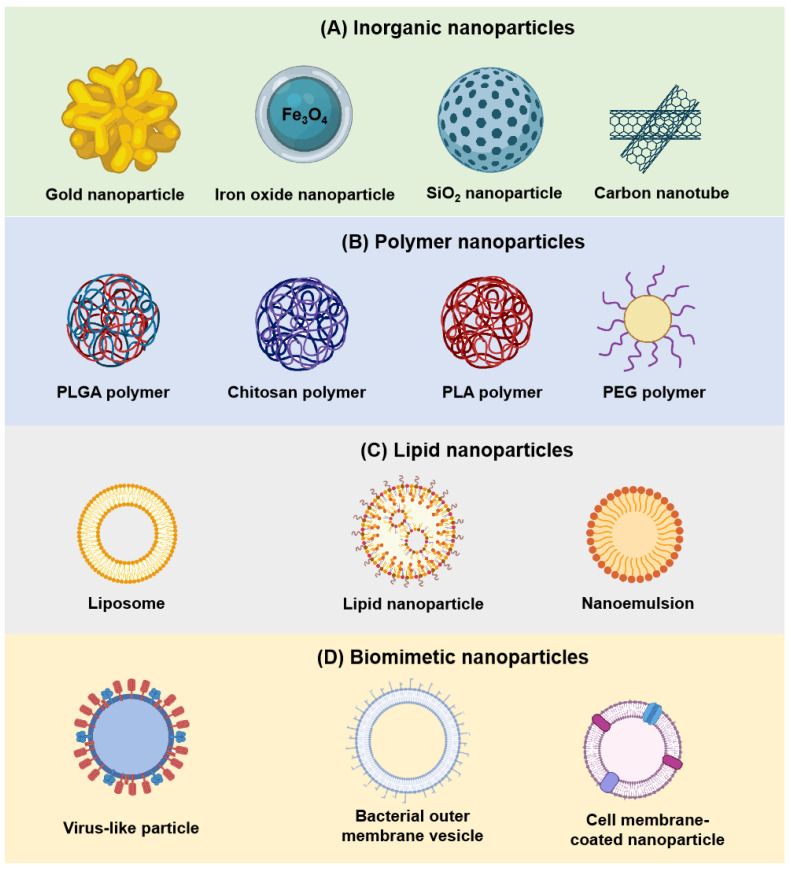
Classes of nanoparticles (NPs) for vaccines. Each class of NP comprises multiple subclasses, with representative examples highlighted herein. (**A**) Inorganic nanoparticles. (**B**) Polymeric nanoparticles. (**C**) Lipid nanoparticles. (**D**) Biomimetic nanoparticles. Created in https://BioRender.com.

**Figure 2 vaccines-13-00900-f002:**
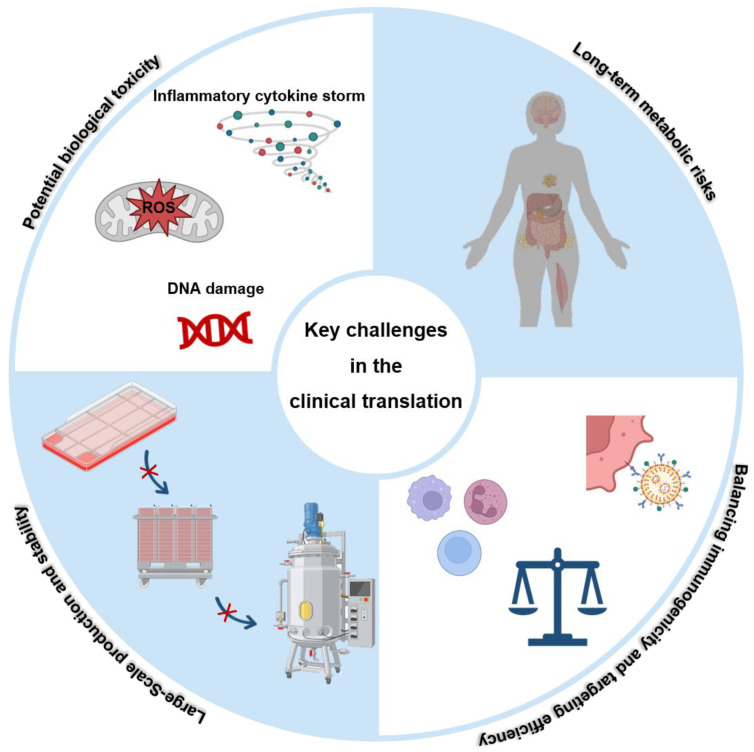
Key challenges in the clinical translation of nanovaccines. Created in https://BioRender.com.

**Table 1 vaccines-13-00900-t001:** Comparison of major nanovaccine delivery systems.

Type of Nanocarrier	Representative Materials	Main Advantages	Main Limitations	Typical Applications	Ref.
Inorganic nanoparticles	Gold NPs iron oxide NPs silica NPs CNTs	High stability Intrinsic immunostimulation EPR effect for passive targeting	Poor biodegradability Potential long-term toxicity and inflammation	Cancer vaccines adjuvant systems	[24,25]
Polymeric nanoparticles	PLGA chitosan PLA, PEG	Controlled and sustained release Biodegradable, FDA-approved Tunable size and surface properties	Possible burst release Process-dependent variability	Infectious diseases, cancer vaccines	[38]
Lipid nanoparticles	Liposomes NLCs SLNs	Excellent biocompatibility Encapsulate hydrophilic and hydrophobic antigens Scalable production	Stability issues (e.g., mRNA vaccines need ultra-cold storage) Limited loading for large biomolecules	mRNA vaccines peptide vaccines	[36]
Biomimetic nanomaterials	VLPs OMVs RBC/DC membranes	Mimic natural pathogens/cells Strong targeting and immunogenicity Versatile surface modification	Complex manufacturing High cost Scale-up challenges	Cancer immunotherapy bacterial and viral vaccines	[3,44]
Self-assembled nanoparticles	Ferritin mi3 E2 encapsulins peptide nanofibers	High antigen density Avoid synthetic carriers Tunable structure and low toxicity	Often early-stage, limited clinical data Sensitive to conditions	Viral vaccines (HPV, HBV, malaria) novel platforms	[66,70,71]

**Table 2 vaccines-13-00900-t002:** Key breakthroughs by disease area (cancer, infectious diseases, autoimmune/allergies) with representative nanovaccine strategies, mechanisms, and clinical prospects.

Disease Area	Representative Nanovaccine Strategies	Mechanisms and Highlights	Clinical Prospects	Disease	Clinical Phase	Refs.
Cancer immunotherapy	Tumor antigen-loaded nanoparticles (lipid, polymer, OMVs) OMV + photothermal combo Nanovaccines + immune checkpoint inhibitors	Enhance DC activation and antigen presentation Reverse immunosuppressive TME Stimulate CTL responses and memory	Promising in melanoma, HCC, breast cancer Combined therapies expected to improve patient survival rates	Pancreatic Neoplasms Melanoma, Ovarian Cancer, Breast cancer Lung cancer, Cervical cancer	Early Phase 1 Phase 1 Phase 2/Phase 3	[83,84,91,109]
Infectious diseases	mRNA-LNP vaccines (Pfizer/BioNTech, Moderna for COVID-19) Multivalent nanoparticles for HIV PLGA, ferritin and chitosan NP vaccines for animal diseases	Enable rapid, stable delivery of nucleic acids and proteins Mimic viral structures to enhance B cell and T cell activation Promote polyantigen presentation and lymph node targeting	Revolutionized pandemic response (COVID-19) Accelerating HIV vaccine pipelines Strengthening veterinary vaccines to combat zoonoses	HIV-1, Epstein–Barr Virus, Ebola, Hepatitis A, Crimean–Congo Hemorrhagic Fever Respiratory Syncytial Virus (RSV) Influenza A, COVID-19, Hepatitis B	Phase 1 Phase 2 Phase 3	[66,96,99,102]
Autoimmune diseases and allergies	LNP delivery of immunoregulatory mRNA (e.g., PD-L1 mRNA) Nanocarriers with tolerogenic or Th2-modulating payloads	Induce tol-APCs to suppress pathogenic T cells and expand Tregs Inhibit DC maturation, Th2 polarization, and allergy-related cytokine release	New precision approaches for RA, colitis, asthma, and allergy Reduce systemic side effects, promising translational potential	-	-	[106,108]

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
