# Peer review of "Nanovaccines: Innovative Advances from Design Strategies to Clinical Translation"

_vaccines, 2025, doi:10.3390/vaccines13090900_

Round 1
Reviewer 1 Report
Comments and Suggestions for Authors
The authors are encouraged to cite as many of the original research articles as possible rather than cite just other review articles. Generalized statement (line 12-13) seem to imply all nanovaccine types will be able to quickly adapt to viral mutations when in fact it is the mRNA and not the LNP that allows for this flexibility.
There is some lack in the precision of language used in the article. For example:
Line 51, not sure how a nanovaccine "targets the immune system". Can the authors expand or explain what this means.
Line 56: While the nanovaccine delivery modality may enhance delivery or ativation of the immune cells, they do not change the "immunogenicity" of the antigen as that is inherent to the immunogen itself and the host's immune repertoire/potential.
Lines 202-203: these cell surface receptors are not unique to DCs.
Lines 241 -244. DNA does not activate STING. Cytosolic DNA may activate AIM2 or cGAS, which in turen generates cyclic dinucleotides that can then activate STING.
Lines 264-266. Why does size matter. Not sure reference 47 addresses this point.
lines 331-332. why do nanoscale dimensions effect "targeting" of tumor cells and why does this enhance an a better anti-tumor immune response. Which of the cited references (67-69) addresses this statement.
Lines 394-395. Other immune responses may be disrupted other than just allergic responses.
lines 490-492. Given the the TME is not an immune induction cite, what is the merit of this approach of vaccine design?
Line 507. Comment regarding universal vaccines. While the reviewer agrees that this is a goal of vaccine research, this might only be achieved if vaccines also target the induction of T cell responses. However, this type of approach may require a change in countries vaccine policy to target an immune mechanism other than antibody (i.e., something that is much less amenable to testing). This is just a comment to the authors.
Line 513: Might be good to list the milestones mentioned.
Reviewer 2 Report
Comments and Suggestions for Authors
The submitted MS entitled: Nanovaccines: Innovative Advances from Design Strategies to Clinical Translation explored the design of nanovaccines, highlighting their applications in cancer, infectious and autoimmune diseases.
The manuscript addresses the topics objectively and clearly, is well written and structured the sub-items appropriate for the proposal, the conclusions were supported by the evidences and arguments presented, and the references were appropriated. Additionally, the subject explored deserves interest and is completely within the journal scope.
I have noted similarities between this review and the following peer-reviewed publication: https://doi.org/10.1021/acsnano.4c15765. However, the authors effectively differentiate the current review by the way they approached the nanovaccines design and mainly because they present and discuss the breakthroughs in the field of application considering cancer immunotherapy, prevention and control of infectious diseases and autoimmune diseases and allergies. Thereby addressing a gap in the existing literature in this regard.
However, considering the central objective of the review and, consequently, its most interesting point, which is to critically examine the existing technical challenges and translational barriers, providing an integrative reference to guide future research and developments in this field, the below issues need to be addressed to strengthen the scientific rigor, and clinical relevance of the review:
1) I consider that it is essential to present and discuss the clinical trials involving nanovaccines within the context addressed. Therefore, I strongly suggest the inclusion and discussion of clinical trials, which complement the present proposal.
2) Another point is that in review manuscripts the presence of figures that show the subject or part of it, addressed in a didactic way, are always very welcome. So, I miss it in the presented proposal.
Because of that I suggest that the MS should be revised considering the suggestions, for further revision.
Reviewer 3 Report
Comments and Suggestions for Authors
This is a small and nice review. Many sections need to be broadened and several references must be added (a lot of statements are made without using references). It seems to be more of a commentary than a review.
My major comments are:
A subsection must be added in terms of the current ignorance shown in the scientific community in terms of nanoparticles stability and functionalization (for targeting purposes for example). Most of the people working with nanovaccines are unaware of steric and electrostatic stability of nanoparticles. Moreover, targeting is nice but the nanovaccine stability might be compromised if people continues to not understand nanoparticle stability. This has to be mentioned in this review. Another section dealing with physisorption and chemisorption must also be included for antigen binding or functionalization purposes.
Other specific comments.
Page 1, line 23-29, the paragraph only mentions viral infections, it must also mention what was summarized in the abstract: cancer immunotherapy.
Introduction, this section must also mention lipid nanoparticles when describing the different types of vaccines, especially those based on RNA (against COVID-19). What are their advantages and disadvantages? This technology already uses a nanovaccine with a practical application.
Table 1, complement this table adding a column with the topic: cost (you only mention this aspect with biomimetic nanoparticles, include all materials). Moreover, add a column or comment about scalability (you only mention this for lipid nanoparticles).
Page 8, line 340-342, "Photothermal therapy locally heats tumor tissues via light-activated agents, inducing cancer cell death and releasing tumor antigens", this seems out of place, are we talking about the same nanosystem (OMV-Mal)?
Section 3.2, this section is not necessary, it has repetitive information. The info about lipid nanoparticles and mRNA has to be moved to the introduction section.
Section 3.3, this section is never mentioned in the abstract or the introduction of this review.
Table 2, besides vaccines from Pfizer and Moderna, a column must be added to indicate which disease area has prototypes being evaluated in clinical trials (as key examples).
Reviewer 4 Report
Comments and Suggestions for Authors
This review explores nonvaccine design and applications in cancer and infectious diseases, with coverage of hurdles in moving towards clinical use. It is well written and interesting to read. Abstract line 11: “personalized nanovaccine designs facilitate precise targeting of cancer cells” is a bit misleading as it indicates that cancer is targeted by the nano construct. Rather, cancer is a downstream target by activated immune cells. Introduction, line 50: Many nanovaccines are not restricted to the 20-100 nm size (most liposomes are larger), but rather have nano-sized features with functional properties. Line 53: it is not antigens that enhance uptake but rather adjuvants, such as PAMPs. Most nanoparticles are rapidly internalized by myeloid cells based on their inherent properties (i.e. being foreign objects). Line 201: Are there references that support the adding PEG improves overall targeting efficiency? Line 469: is targeted delivery a goal for nanovaccines? The targets are myeloid cells, which rapidly take up NPs. Similarly, is EPR (line 88) real and advantageous for nanovaccines? Accumulation at pathological sites is usually by myeloid uptake and trafficking to sites of inflammation.
Reviewer 5 Report
Comments and Suggestions for Authors
The review “Nanovaccines: Innovative Advances from Design Strategies to Clinical Translation” (vaccines-3794166) provides a detailed and well-organized summary of nanovaccines. It covers design, materials, functionalization, and applications in cancer immunotherapy, infectious diseases, and autoimmune disorders. The review also discusses challenges in clinical translation. The topic is timely and relevant, especially after the COVID-19 pandemic. It combines materials science and immunology, giving value to both researchers and clinicians. With minor revisions, it can be even better.
- The abstract could be shorter. This would improve readability and highlight key new insights.
- Some sections, like functional modifications and adjuvant synergy, contain heavy technical jargon. Simplifying these and briefly explaining terms like "vesicular stomatitis virus glycoprotein (VSVG)" would improve accessibility.
- Adding figures could help too.
- Including schematic diagrams showing nanovaccine design, targeting and immune activation mechanisms, and nanocarrier types would strengthen the manuscript.
- The clinical translation section is solid but could benefit from more detailed case studies or examples beyond COVID-19 mRNA vaccines.
- The section on biological toxicities is thorough. However, distinguishing toxicity profiles for different nanomaterials and ways to reduce risks would be useful.
- There are some typographical errors and inconsistent spacing, likely from proofreading. Some sentences are incomplete.
- Overall, tightening some sentences would improve flow of the review
Round 2
Reviewer 2 Report
Comments and Suggestions for Authors
As already mentioned in the first review, the submitted MS, that explored the design of nanovaccines, highlighting their applications in cancer, infectious and autoimmune diseases, addresses the topics objectively and clearly, is well written and structured the sub-items appropriate for the proposal, the conclusions were supported by the evidences and arguments presented, and the references were appropriated.
The revised version covered all the suggestions, such as, the inclusion nanovaccines clinical trial at Table 2, as well as, the inclusion of Figures 1 and 2, to clearly present the topic which improve its readability.
So, considering that and the fact that the explored subject deserves interest and is completely within the journal scope, I recommend its acceptance in the present form.
Reviewer 3 Report
Comments and Suggestions for Authors
The authors have successfully addressed all my previous comments.